# Aortic Valve Replacement: Totally Endoscopic versus Mini-Sternotomy

**DOI:** 10.3390/jcm12237300

**Published:** 2023-11-24

**Authors:** Alaaddin Yilmaz, Jade Claessens, Loren Packlé, Silke Van Genechten, Kübra Dönmez, Camille Awouters, Lieven Herbots

**Affiliations:** 1Department of Cardiothoracic Surgery, Jessa Hospital, Stadsomvaart 11, 3500 Hasselt, Belgium; alaaddin.yilmaz@jessazh.be (A.Y.); loren.packle@jessazh.be (L.P.); silke.vangenechten@jessazh.be (S.V.G.); camille.awouters@jessazh.be (C.A.); 2Department of Medicine and Life Sciences, Universiteit Hasselt, Martelarenlaan 42, 3500 Hasselt, Belgium; kubra.donmez@student.uhasselt.be (K.D.); lieven.herbots@jessazh.be (L.H.); 3Department of Cardiology, Jessa Hospital, Stadsomvaart 11, 3500 Hasselt, Belgium

**Keywords:** minimally invasive cardiac surgery, aortic valve replacement, mini-sternotomy, totally endoscopic surgery

## Abstract

(1) Background: The development of totally endoscopic aortic valve replacement has the potential to enhance clinical results compared to mini-sternotomy. To our knowledge, no comparison between these two techniques has been conducted before. Therefore, the objective of this retrospective study is to examine the results after both totally endoscopic and mini-sternotomy approaches. (2) Methods: This study covered all elective patients who underwent isolated aortic valve replacement, either totally endoscopically (n = 392) or through a mini-sternotomy (n = 323), between 2013 and 2021. Multivariable analysis was used to account for baseline variations between the two groups. All data were retrospectively gathered and analysed. The primary objective of this study was the one-year mortality rate. (3) Results: The mean aortic cross-clamping and cardiopulmonary bypass times were significantly longer in the totally endoscopic approach (cross-clamping: 43.73 ± 13.71 min and 61.93 ± 16.76 min, *p*-value < 0.001; CPB time: 64.86 ± 23.02 min and 93.23 ± 23.67 min, *p*-value < 0.001). However, perioperative bleeding was lower (706.40 ± 542.77 mL and 444.50 ± 515.84 mL, *p*-value < 0.001). The primary objective, one-year survival, did not significantly differ between both groups (Mini-AVR: 94.5% vs TEAVR: 93.3%, *p*-value = 0.520). (4) Conclusions: Our results show that totally endoscopic aortic valve replacement has comparable clinical results compared to aortic valve replacement through mini-sternotomy.

## 1. Introduction

The ultimate goal of research in the field of cardiac surgery is to improve functional outcomes such as postoperative recovery, blood loss, intensive care unit (ICU) stay, pain, and cosmetic outcomes. Many cardiac centres today provide surgery by means of minimally invasive access, such as partial upper sternotomies (J- vs. inverse T-shaped) and anterolateral thoracotomies (right anterolateral thoracotomy (RALT) vs. right anterior thoracotomy (RAT)) [1,2]. In accordance with this evolution, the development of totally endoscopic techniques might have the potential to improve clinical outcomes. Regarding totally endoscopic aortic valve replacement (TEAVR), previous studies have indicated that it is a safe and feasible alternative to perform AVR [3,4,5,6]. In the field of ventricular septal defect repair, a comparison between totally endoscopic and Mini-sternotomy-AVR (Mini-AVR) showed that outcomes were similar in both groups [7]. However, to our knowledge, no studies compared the clinical outcomes after TEAVR to Mini-AVR. Elective AVR has been carried out through a mini-sternotomy in our facility since 2005, followed by TEAVR in October 2017. This monocentric retrospective study aims to compare the clinical outcomes after TEAVR to Mini-AVR.

## 2. Materials and Methods

### 2.1. Patients

All elective TEAVR patients and mini-AVR patients who underwent surgery between 2013 and 2021 were included in this study. The first TEAVR was performed at the study institution in October 2017. All patients before this time point underwent a mini-AVR (Figure 1). Between October 2017 and December 2019, both procedures were done based on the surgeon’s availability. All TEAVRs were performed by one surgeon, while the Mini-AVRs were done by two surgeons. No other patient selection was made. The exclusion criteria were previous cardiac surgery and concomitant surgeries.

### 2.2. Surgical Technique

Both the Mini-AVR and TEAVR surgical techniques were previously described, and a video of the TEAVR technique is also available [8,9]. The Mini-AVR is performed through a J-shaped partial upper sternotomy ending in the third intercostal space. In the case of TEAVR, aortic access is gained by a 1.5 to 2 cm working port in the 2nd midclavicular intercostal space, combined with three 5 mm trocars in the 2nd and 3rd intercostal spaces. The whole procedure is performed in a totally endoscopic fashion using only endoscopic instruments and zero-degree optics. Tips and tricks regarding this technique are published by Yilmaz et al. [9].

### 2.3. Outcomes

The primary objective of this study was to assess the one-year all-cause mortality rate. The composite endpoint of major adverse cardiac and cerebrovascular events (MACCEs), including myocardial infarction, cerebrovascular accident (CVA), prosthesis deterioration, and cardiac mortality, was the secondary objective of this study. Prosthetic deterioration is defined as degeneration and/or hemodynamic dysfunction caused by intrinsic irreversible alterations to the prosthetic valve, such as calcification, leaflet fibrosis, tears, or flail [10]. Collected perioperative parameters consisted of clamping and cardiopulmonary bypass (CPB) times, valve characteristics, and bleeding. Perioperative bleeding is the volume (mL) of blood drawn by a cell saver (Sorin S.p.A., Mirandola, Italy) suction during surgery. In contrast, postoperative bleeding is gauged 24 h later through the thoracic drains. Other clinical outcomes included surgical reoperations, neurological complications, newly developed atrial fibrillation (AF), permanent pacemaker (PPM) implantation, intensive care unit (ICU), hospital length of stay (LOS), and mortality. Stroke timing was defined by the Valve Academic Research Consortium 3 as acute (≤24 h after the index procedure), sub-acute (>24 h and ≤30 days after the index procedure), early (>30 days and ≤1 year after the index procedure), and late (>1 year after the index procedure). Vascular complications were divided into minor and major categories, according to the Valve Academic Research Consortium 3 [11].

### 2.4. Statistical Analysis

The data were retrospectively collected and analysed. Statistical significance was defined as a *p*-value of <0.05. The normal distribution of the data was evaluated through QQ-plots and the Shapiro–Wilk test. Data are expressed as mean ± standard deviation, numbers (n), and frequencies (%). 

Moreover, to correct the baseline differences between the two groups, all possible confounders were determined based on demographic variables that were significantly different. These confounders included age, smoking, chronic kidney disease (CKD), AF, arterial hypertension, New York Heart Association (NYHA) score, cardiac family history, preoperative neurological incidents, and the presence of a bicuspid valve. Postoperative variables were considered as the outcome in a linear or logistic regression model, using the abovementioned covariates. The odds ratio or estimate and the corresponding 95% confidence intervals (CIs) were reported. Differences between the two groups were compared using a chi-squared test, Fischer’s exact test, and unpaired t-test, as appropriate. Survival was evaluated using a Kaplan—Meier analysis for mortality and cumulative incidence function for MACCE. Data analysis was performed using an intention-to-treat principle using R Core Team (2021) R: a language and environment for statistical computing. R Foundation for Statistical Computing, Vienna, Austria.

## 3. Results

### 3.1. Patient Demographics

Between January 2013 and December 2021, 715 patients were referred for an AVR. Among these, 323 patients underwent Mini-AVR, while 392 underwent TEAVR. The mean follow-up was 2081 ± 895.43 days for the Mini-AVR patients, with a follow-up index of 0.91 ± 0.21. The TEAVR patients had a mean follow-up period of 742.9 ± 519.11 days, with a follow-up index of 0.75 ± 0.34. Patient demographics are displayed in Table 1.

### 3.2. Perioperative Data

The mean aortic cross-clamping and CPB times were significantly longer in TEAVR (cross-clamping: 43.74 ± 13.73 min and 61.93 ± 16.76 min, *p*-value < 0.001; CPB time: 64.86 ± 23.02 min and 93.23 ± 23.67 min, *p*-value < 0.001). However, there was significantly lower perioperative bleeding in the TEAVR group (706.40 ± 542.77 mL and 444.50 ± 515.84 mL, *p*-value < 0.001). The need for packed cells, plasma, or platelet transfusion was significantly lower in TEAVR patients. The most common valve prosthesis was Epic™ (St. Jude Medical) in both groups. However, in the TEAVR group, sutureless valves were implanted to a significantly higher extent (TEAVR: 18.37%; Mini-AVR: 1.86%; *p*-value < 0.001). All perioperative data are presented in Table 2.

### 3.3. Postoperative Parameters 

Although there was significantly lower perioperative bleeding, the 24 h bleeding was significantly higher in the TEAVR group (230.60 ± 159.04 mL and 297.30 ± 291.52 mL, *p*-value < 0.001, 95% CI [33.23; 123.53]). However, the need for postoperative transfusions was still significantly higher in the Mini-AVR group (*p* < 0.001, *p =* 0.002, and *p* < 0.001 for packed cells, fresh frozen plasma, and platelets, respectively). Additionally, the ventilation time and ICU length of stay (LOS) were similar in both groups. Moreover, the occurrence of new-onset AF and the need for electric cardioversion were significantly lower in the TEAVR group (AF: 32.41% and 24.74%, *p*-value = 0.025, OR:0.61, 95% CI [0.39 to 0.95]; electric cardioversion: 14.81% and 5.61%, *p*-value < 0.001, OR: 0.58, 95% CI [0.29 to 1.14]). Furthermore, within 48 h, a reoperation was needed in 3.09% of the Mini-AVR patients, while 4.08% of the TEAVR patients underwent an early reoperation (*p*-value = 0.206, OR: 1.93, 95% CI [0.005 to 0.15]). In the case of TEAVR patients, this reoperation was also performed totally endoscopically. A reoperation within one week was needed in fewer TEAVR patients (Mini-AVR: 3.09%; TEAVR: 1.79%; *p*-value = 0.102, OR: 0.14, 95% CI [0.01 to 1.39]). The implantation of PPM was significantly higher in the TEAVR group. A PPM was implanted in 3.70% of Mini-AVR patients, while 8.16% of TEAVR patients needed a PPM (*p*-value < 0.001, OR: 4.29, 95% CI [1.80 to 10.80]). In contrast, neurological complications were significantly lower in the TEAVR group. A CVA occurred in significantly more Mini-AVR patients, 4.01% of the Mini-AVR patients, and 2.81% of TEAVR patients (*p*-value = 0.0.26). In 8.95% of the Mini-AVRs, acute kidney injury (AKI) developed, while it occurred in only 3.06% of the TEAVR patients (*p*-value = 0.464, OR: 0.73, 95% CI [0.30 to 1.67]). The need for renal replacement therapy was similar in both groups (0.62% and 2.04%, *p*-value = 0.365, OR: 2.48, 95% CI [0.41 to 22.97]). Lastly, the mean hospital LOS was significantly shorter in the TEAVR group (10.12 ± 7.63 days and 7.09 ± 10.96 days, *p*-value = 0.002, estimate: −2.78, 95% CI [−4.56 to −0.99]). 

During the follow-up, a reoperation was needed in 3.09% of Mini-AVR patients and in 1.79% of TEAVR patients (*p*-value = 0.204, OR: 0.46, 95% CI [0.13 to 1.49]). Moreover, there were fewer paravalvular leakage, endocarditis, and cardiac readmissions after TEAVR (*p*-value = 0.105, OR: 0.44, 95% CI [0.16 to 1.19]; *p*-value = 0.357, OR: 0.57, 95% CI [0.16 to 1.84]; *p*-value = 0.200, OR: 0.74, 95% CI [0.46 to 1.18], respectively). The postoperative parameters are displayed in Table 3. 

### 3.4. MACCE

Within 30 days, a MACCE occurred in six patients (1.86%) undergoing a Mini-AVR and in eight patients (2.04%) undergoing TEAVR (*p*-value = 0.830). Seven more Mini-AVR patients (2.17%) and four more TEAVR patients (1.02%) developed MACCE over the course of a year (*p*-value = 0.412). During the three-year follow-up, freedom from MACCE was 88.2% for the Mini-AVR patients and 91.1% for the TEAVR patients (*p*-value = 0.400). Seven additional Mini-AVR patients (2.17%; Figure 2) experienced MACCE during this period.

### 3.5. All-Cause Mortality 

One TEAVR patient (0.26%) passed away during surgery, whereas no one in the mini-AVR group did (*p* = 0.999). Within 30 days, six additional TEAVR patients (1.53%) and four mini-AVR patients (1.24%) deceased (*p* = 0.826). One year after the surgery, the survival rate for Mini-AVR patients was 94.5%. This rate is not significantly different from the 93.3% rate in TEAVR patients (*p*-value = 0.520). Moreover, during the three-year period, 89.9% of the Mini-AVRs and 85.6% of the TEAVR patients survived (*p*-value = 0.150); Figure 3 Table 4).

## 4. Discussion

This retrospective study aimed to compare the results of two minimally invasive procedures: Mini-AVR and TEAVR. Regarding the primary objective, one-year survival, no significant difference was observed between Mini-AVR and TEAVR (94.5% vs 93.3%, *p*-value = 0.500). Furthermore, the 30-day mortality was 1.24% after Mini-AVR and 1.53% after TEAVR, which is lower compared to a study that investigated Mini-AVR and AVR via a right anterior thoracotomy (RAT) (2.3% and 3.1%, respectively) [12]. In another study that compared Mini-AVR to RAT, similar 30-day mortality rates were seen (1.2% and 2.8%, respectively) [2]. Additionally, the three-year survival did not significantly differ between the two groups.

No difference in MACCE between Mini-AVR and TEAVR was observed in our series. To our knowledge, no other study compares Mini-AVR with TEAVR, which is why a direct comparison is impossible. However, in a previous study comparing trans-right axillary (TAX) AVR with conventional AVR, 30-day MACCE was 3.7% in the TAX-AVR group, which is higher than the 1.86% in the Mini-AVR and 2.04% in the TEAVR group [13].

The transition to minimally invasive procedures, especially endoscopic techniques, has always come with longer aortic clamping and CPB times caused by growing expertise, inadequate equipment, and technically more challenging approaches [14,15]. Our study confirmed that the aortic clamping and CPB times were significantly longer in the TEAVR group compared to Mini-AVR (cross-clamping: 43.74 ± 13.73 min and 61.93 ± 16.76 min, *p*-value < 0.001; CPB time: 64.86 ± 23.02 min and 93.23 ± 23.67 min, *p*-value < 0.001). Nevertheless, these OR times are still in accordance with the accepted times for an isolated AVR by sternotomy [13]. Additionally, we have similar results compared to Mourad et al. (cross-clamping: Mini-AVR: 63.61 ± 16.115 min vs RAT: 70.75±33.274 min; CPB: Mini-AVR: 91.90 ± 26.365 min vs RAT: 112.24 ± 51.634 min) and Glauber et al. (RAT; cross-clamping: 97 ± 29 min; CPB: 134 ± 47 min) [12,16].

The perioperative bleeding was significantly lower in the TEAVR group, and significantly fewer transfusions were needed. These results align with Mourad et al. where there was 267.95 ± 65.18 mL blood loss in the Mini-AVR group and 251.3 8 ± 58.76 mL in the RAT group. Hancock et al. reported less blood loss after Mini-AVR (181.6 ± 138.7 mL), but we observed less blood loss compared to a sternotomy (306.9 ± 348.6 mL) [17]. Moreover, thoracoscopic reoperation for bleeding, tamponade, or hemothorax occurred in the same amount in both Mini-AVR and TEAVR. Reoperation surgery is always performed endoscopically in TEAVR patients, while in only one of the mini-AVR cases the reoperation was done endoscopically. The percentage of reoperations after TEAVR (3.32%) is slightly lower or similar compared to RAT (8.6% and 5.1%) [12,16].

Likewise, the ventilation time and ICU LOS were not significantly different between both groups but were longer in our series compared to other trials [2,12,16]. However, our hospital does not have a median intensive care unit. Patients are directly transferred to the regular ward after their stay in the ICU. The hospital LOS, on the other hand, was significantly lower after TEAVR compared to Mini-AVR (10.12 ± 7.63 days and 7.09 ± 10.96 days, *p*-value = 0.002, estimate: −2.78, 95% CI [−4.56 to −0.99]). In comparison with RAT, the hospital LOS was lower or similar in TEAVR patients and also in line with other techniques (8.7 days, 7.4 days, 9.75 ± 2.51 days and 6 days) [2,5,12,16]. Regarding ICU and hospital LOS, differences in government payment systems should be considered. In the Belgian healthcare system, high bed occupancy is financially favourable for the hospital.

PPM implantation rates were significantly different between the groups. This PPM rate can be explained by the fact that there were significantly more sutureless valve prostheses in the TEAVR group. Sutureless valve prostheses are associated with a higher PPM implantation rate [18]. In 9 of the 12 PPM in the TEAVR group, a sutureless valve was implanted. This indicates that the high PPM rate in the TEAVR group is most likely related to the use of sutureless valves and not the technique itself. Moreover, PPM implantation was required in more patients in our series compared to RAT [2,12,16]. It should be noted that the threshold for PPM implantation is low in our hospital. Compared to the results of transcatheter pacemaker implantation in low-risk patients, where pacemaker implantation was required in 6.6% of patients and conventional AVR (2–7%,) the results of this study are acceptable [19,20,21].

Neurological complications are low in both the Mini-AVR and TEAVR patients but occurred to a significantly greater extent in the Mini-AVR group. De-airing the left ventricle before closing the aortotomy and CO_2_ insufflation in the operating field are crucial to reducing perioperative stroke risk. To prevent an overload of high-flow air reaching the pulmonary venous system, CO_2_ insufflation must remain at a low continuous flow. When ventilation is started at the end of the procedure, trapped air bubbles can be released, and emboli may occur. Therefore, the left ventricular vent is only removed after filling the ventricle combined with ventilation to ensure all macroscopic air bubbles have vanished.

Overall, the advantages of TEAVR over Mini-AVR are less surgical damage and bleeding, no sternum infections, fewer neurological complications, and a shorter hospital LOS. Given our promising results with TEAVR and Mini-AVR, a stronger transition to a less invasive technique by the surgical community is justified, despite the fact that TEAVR is currently not a widespread technique, with limited knowledge [9,22,23,24].

### Limitations

This report’s retrospective nature is a limitation that leads to selection bias. We tried solving this to correct for baseline differences using linear or logistic regression models. The external validity of this trial was also constrained by the fact that just one surgeon performed TEAVR. The reproducibility of this technique should be demonstrated in a multicenter study. Additionally, the two techniques were not performed during the same time period. Since TEAVR was developed at a later stage, differences could be caused by improved perioperative management and postoperative patient care.

## 5. Conclusions

Our findings demonstrate that TEAVR has similar clinical outcomes to Mini-AVR. Although aortic cross-clamping and CPB times are significantly longer in TEAVR, and the need for PPM is higher, these patients experience significantly lower perioperative bleeding and a reduced need for blood transfusions. Additionally, there are fewer neurological complications, and these patients remained at the hospital for a shorter period.

In conclusion, this retrospective analysis, comparing TEAVR and Mini-AVR in 715 consecutive patients, demonstrates satisfactory results for TEAVR, with acceptable mortality and morbidity rates.

## Figures and Tables

**Figure 1 jcm-12-07300-f001:**
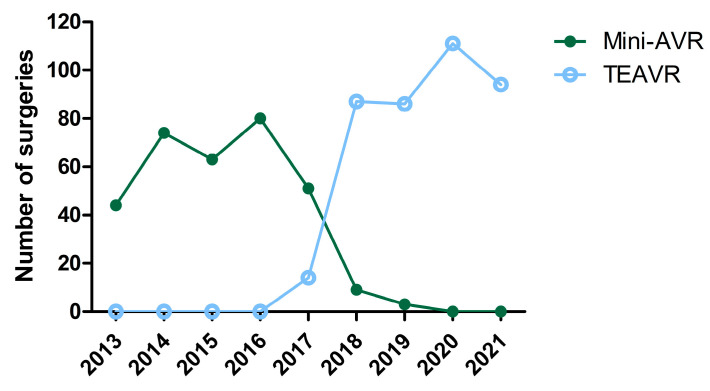
Number of aortic valve replacements through mini-sternotomy (mini-AVR; n = 323) and totally endoscopic (TEAVR; n = 392) techniques from 2013 until 2021.

**Figure 2 jcm-12-07300-f002:**
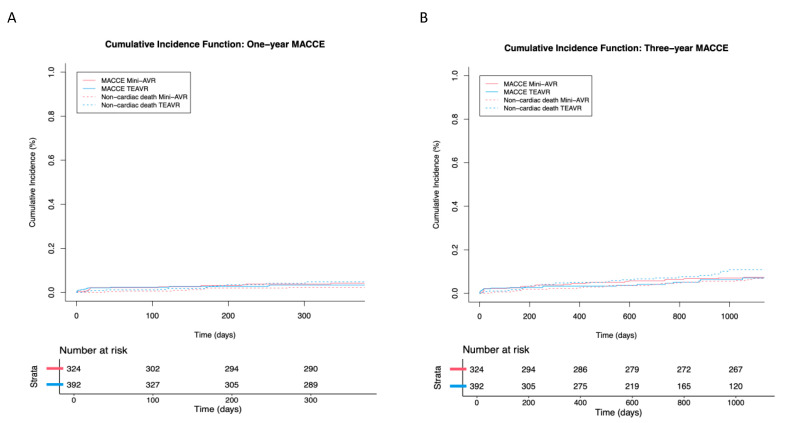
Estimated survival of major adverse cardiac and cerebrovascular events (MACCE; cumulative incidence function) for aortic valve replacement through mini-sternotomy (mini-AVR; n = 323) and totally endoscopic (TEAVR, n = 392) techniques after one year (**A**) and three years (**B**).

**Figure 3 jcm-12-07300-f003:**
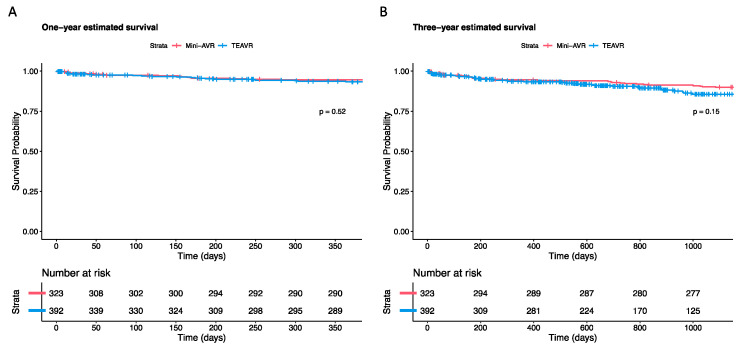
One year (**A**) and three years (**B**) estimated survival of all-cause mortality (Kaplan—Meier) after aortic valve replacement through mini-sternotomy (mini-AVR; n = 323) and a totally endoscopic (TEAVR; n = 392) technique.

**Table 1 jcm-12-07300-t001:** Patient demographics.

	Mini-AVR (N = 323)	TEAVR (N = 392)	*p*-Value
Age (years)	72.65 ± 9.93	71.20 ± 10.85)	0.063
Octogenarians	86 (26.63)	96 (24.49)	0.514
Gender (male)	179 (55.42)	234 (59.69)	0.249
BMI (kg/m^2^)	27.89 ± 4.68	27.57 ± 4.74	0.373
EuroSCORE II	2.54 ± 2.22	2.33 ± 2.24	0.251
NYHA score			0.015
• I • II • III • IV	98 (30.34) 157 (48.61) 62 (19.20) 2 (0.62)	124 (31.63) 220 (56.12) 43 (10.97) 4 (1.02)	
Comorbidities • Smoking - Active - Stopped • Diabetes mellitus - type I - type II	45 (13.93) 105 (32.51) 4 (1.24) 78 (24.15)	78 (19.90) 60 (15.31) 3 (0.77) 84 (21.43)	<0.001 0.544
• Arterial hypertension • Dyslipidemia • Family cardiac history • Atrial fibrillation • Pacemaker • Conduction disturbance • Chronic kidney disease	223 (69.04) 204 (63.16) 95 (29.41) 72 (22.29) 11 (3.41) 38 (11.76) 58 (17.96)	238 (60.71) 257 (65.56) 76 (19.39) 61 (15.56) 13 (3.32) 26 (6.63) 270 (68.88)	0.019 0.504 0.002 0.021 0.661 0.018 <0.001
History of:			0.013
• CVA • TIA	20 (6.19) 8 (2.48)	12 (3.06) 12 (3.06)	
LVEF (%)	58.12 ± 11.92	58.83 ± 35.82	0.9187
TTE • Peak aortic valve gradient (mm Hg) • Mean aortic valve gradient (mm Hg) • AVA (cm^2^)	80.07 ± 23.48 50.86 ± 16.13 0.74 ± 0.21	73.2 ± 21.95 48.04 ± 15.78 0.78 ± 0.21	0.002 0.090 0.050
Bicuspid valve	30 (9.29)	72 (18.37)	<0.001

Data presented as mean ± standard deviation or n (%). AVA: aortic valve area; BMI: body mass index; CKD: chronic kidney disease; CVA: cerebrovascular accident; Euroscore II: European System for Cardiac Operative Risk Evaluation; LVEF: left ventricular ejection fraction; Mini-AVR: aortic valve replacement through mini-sternotomy; NYHA: New York Heart Association; SMD: standardised mean difference: TEAVR: totally endoscopic aortic valve replacement; TIA: transient ischemic attack; TTE: transthoracic echocardiography.

**Table 2 jcm-12-07300-t002:** Intraoperative parameters.

	Mini-AVR (N = 323)	TEAVR (N = 392)	*p*-Value
Indication for surgery			0.202
• Aortic stenosis • Aortic regurgitation • Aortic stenosis + regurgitation	289 (89.47) 10 (3.10) 24 (7.43)	357 (91.07) 17 (4.34) 18 (4.59)	
Cross-clamping time (h)	43.74 ± 13.73	61.93 ± 16.76	<0.001
CPB time (h)	64.86 ± 23.02	93.23 ± 23.67	<0.001
Unplanned mechanical circulatory support	0 (0)	3 (0.77)	0.115
Perioperative bleeding (mL)	706.40 ± 542.77	444.50 ± 515.84	<0.001
Transfusion • PC • Amount PC • FFP • Amount FFP • Platelets • Amount platelets	111 (34.37) 2.03 ± 1.19 16 (4.95) 2.94 ± 1.12) 28 (8.67) 1.21 ± 0.42	57 (14.54) 1.84 ± 0.90 1 (0.26) 3 ± 0 1 (0.26) 1 ± 0	<0.001 <0.001 <0.001
Aortic prosthesis			<0.001
• Epic™ (St.Jude Medical) • Trifecta™ (St.Jude Medical) • Avalvus (Medtronic) • Carbomedics (LivaNova) • Perceval (LivaNova) • Magna ease (Edwards Lifesciences) • Other	254 (78.64) 50 (15.48) 0 (0) 9 (2.79) 6 (1.86) 2 (0.62) 2 (0.62)	218 (55.61) 70 (17.86) 9 (2.30) 1 (0.26) 72 (18.37) 14 (3.57) 7 (1.79)	
Conversion			0.288
• Sternotomy • Mini-sternotomy	0 (0) /	0 (0) 1 (0.26)	

Data presented as mean (standard deviation) or n (%). CPB: cardiopulmonary bypass; FFP: fresh frozen plasma; Mini-AVR: aortic valve replacement through mini-sternotomy; PC: packed cells; TEAVR: totally endoscopic aortic valve replacement.

**Table 3 jcm-12-07300-t003:** Postoperative parameters.

Continuous Data					
	Mini-AVR (N = 323)	TEAVR (N = 392)	*p*-Value	Estimate	95% CI
Ventilation time (h)	10.51 ± 18.93	12.85 ± 64.55	0.409	3.90	−5.35–13.16
Bleeding 24 h (mL)	230.60 ± 159.04	297.30 ± 291.52	<0.001	78.38	33.23–123.53
ICU LOS (h)	69.64 ± 68.68	60.41 ± 133.41	0.853	−1.90	−22.08–18.27
Hospital LOS (days)	10.12 ± 7.63	7.09 ± 10.96	0.002	−2.78	−4.56 to −0.99
LVEF (%) • In-hospital • Follow-up	56.39 ± 9.64 55.67 ± 11.37	55.88 ± 10.66 57.90 ± 10.39	0.883 0.295	−0.17 1.64	−2.47–2.12 −1.42–4.69
TTE In-hospital • Peak aortic valve gradient (mm Hg) • Mean aortic valve gradient (mm Hg) • AVA (cm^2^)	22.81 ± 8.31 13.05 ± 4.85 1.76 ± 0.40	20.21 ± 9.27 12.05 ± 6.10 1.93 ± 0.48	0.057 0.371 0.022	−2.38 −0.85 0.22	−4.83–0.07 −2.70–1.00 0.03–0.41
Follow-up • Peak aortic valve gradient (mm Hg) • Mean aortic valve gradient (mm Hg) • AVA (cm^2^) • TTE follow-up (days)	22.63 ± 11.19 14.09 ± 8.12 1.63 ± 0.51 1766 ± 1011.46	20.13 ± 12.19 12.01 ± 5.27 1.77 ± 0.42 605.5 ± 507.95	0.034 0.081 0.605 <0.001	−3.27 −2.16 0.07 −1227.70	−6.27 to −0.26 −4.57–0.25 −0.18–0.31 −1383.26 to −1072.14
Clinical follow-up time (days)	2 081 ± 895.43	742.9 ± 519.11	<0.001	−1381.11	−1511.71 to −1250.50
Clinical follow-up index	0.91 ± 0.21	0.75 ± 0.34	<0.001	−0.20	−0.25 to −0.15
**Categorical Data**					
	**Mini-AVR** **(N = 323)**	**TEAVR** **(N = 392)**	***p*-Value**	**OR**	**95% CI**
Unplanned mechanical circulatory support	3 (0.93)	4 (1.02)	0.077	0.08	0.002–1.03
Transfusions • PC • Amount PC • FFP • Amount FFP • Platelets • Amount platelets	75 (23.15) 3.05 ± 2.33 17 (5.25) 212.12 ± 0.78 21 (6.48) 1.33 ± 0.48	64 (16.33) 3.80 ± 3.05 13 (3.32) 1.85 ± 0.55 20 (5.10) 1.2 ± 0.41	<0.001 0.002 <0.001	0.21 0.05 0.04	0.13–0.36 0.006–0.28 0.005–0.15
Reoperation (<48 h)	10 (3.09)	16 (4.08)	0.206	1.93	0.005–0.15
• Bleeding • Tamponade • Emphysema	4 (1.23) 6 (1.85) 0 (0)	13 (3.32) 5 (1.28) 0 (0)			
Reoperation (>48 h)	3 (0.93)	2 (0.51)	0.102	0.14	0.010–1.39
• Bleeding • Tamponade • Emphysema • Mediastinitis • Irritation steel wires • Pericardial fenestration	0 (0) 1 (0.31) 0 (0) 1 (0.31) 1 (0.31) 0 (0)	0 (0) 1 (0.26) 0 (0) 0 (0) 0 (0) 1 (0.26)			
Reoperation (>7 days)	10 (3.09)	7 (1.79)	0.204	0.46	0.13–1.49
• Late tamponade • Endocarditis • Irritation steel wires • Prosthesis failure • Pericardial fenestration	1 (0.31) 7 (2.16) 1 (0.31) 1 (0.31) 0 (0)	2 (0.51) 3 (0.77) 0 (0) 1 (0.26) 1 (0.26)			
Paravalvular leakage • In-hospital • Follow-up	6 (1.85) 10 (3.09)	2 (0.51) 11 (2.81)	0.624 0.105	0.57 0.44	0.05–4.77 0.16–1.19
Endocarditis	10 (3.09)	6 (1.53)	0.357	0.57	0.16–1.84
Valve mispositioning	0 (0)	0 (0)	-	-	-
Mediastinitis	1 (0.31)	0 (0)	- *	- *	- *
New-onset AF Electric cardioversion	104 (32.10) 48 (14.81)	97 (24.74) 22 (5.61)	0.030 0.118	0.61 0.58	0.39–0.95 0.29–1.14
Conduction disturbance	44 (13.58)	72 (18.37)	0.028	1.79	1.07–3.04
30-day PPM	12 (3.70)	32 (8.16)	0.001	4.29	1.80–10.80
Vascular complications			0.090	0.38	0.12–1.13
• Minor • Major	15 (4.63) 0 (0)	7 (1.79) 1 (0.26)			
Neurological complications	30 (9.26)	20 (5.10)	0.026	0.44	0.21–0.90
• CVA • TIA • Delirium	13 (4.01) 4 (1.23) 13 (4.01)	11 (2.81) 2 (0.51) 7 (1.79)			
Neurological timing			0.246	0.63	0.29–1.41
• Acute • Sub-acute • Early • Late	2 (0.62) 15 (4.63) 4 (1.23) 21 (6.48)	3 (0.77) 11 (2.81) 3 (0.77) 9 (2.30)			
AKI Renal replacement therapy	29 (8.95) 2 (0.62)	12 (3.06) 8 (2.04)	0.464 0.365	0.73 2.48	0.30–1.67 0.41–22.97
Cardiac related readmissions	73 (22.53)	72 (18.37)	0.200	0.74	0.46–1.18

Data presented as mean (standard deviation) or n (%). * Not clinically relevant. AF: atrial fibrillation; AKI: acute kidney injury; AVA: aortic valve area; CVA: cerebrovascular accident; FFP: fresh frozen plasma; ICU: intensive care unit; LOS: length of stay; LVEF: left ventricle ejection fraction; Mini-AVR: aortic valve replacement through mini-sternotomy; PC: packed cells; PPM: permanent pacemaker; SMD: standardised mean difference; TEAVR: totally endoscopic aortic valve replacement; TIA: transient ischemic attack; TTE: transthoracic echocardiography.

**Table 4 jcm-12-07300-t004:** Mortality.

	Mini-AVR (N = 323)	TEAVR (N = 392)	*p*-Value	OR	95% CI
Periprocedural mortality	0 (0)	1 (0.26)	0.999	2.12 × 10 ^24^	0.00–Inf
Causes of death • Cardiovascular • Valve-related • Non-cardiovascular • Not reported	0 (0) 0 (0) 0 (0) 0 (0)	1 (0.26) 0 (0) 0 (0) 0 (0)			
30-day mortality rate	4 (1.24)	7 (1.79)	0.826	1.99	−0.54–8.12
Causes of death • Cardiovascular • Valve-related • Non-cardiovascular • Not reported	4 (1.24) 0 (0) 0 (0) 0 (0)	4 (1.03) 0 (0) 3 (0.77) 0 (0)			
One-year survival (%)	94.5	93.3	0.520 *		
Causes of death • Cardiovascular • Valve-related • Non-cardiovascular • Not reported	6 (1.86) 0 (0) 3 (0.93) 8 (2.48)	5 (1.28) 0 (0) 11 (2.81) 7 (1.79)			
Three-year survival (%)	89.9	85.6	0.150 *		
Causes of death • Cardiovascular • Valve-related • Non-cardiovascular • Not reported	8 (2.48) 0 (0) 8 (2.48) 15 (4.65)	9 (2.30) 0 (0) 17 (4.34) 12 (3.06)			

Data presented as mean (standard deviation), n (%) or %. * Log-rank test of Kaplan—Meier analysis. CVA: cerebrovascular accident; Mini-AVR: aortic valve replacement through mini-sternotomy; TEAVR: totally endoscopic aortic valve replacement.

## Data Availability

The article’s data will be shared upon reasonable request to the corresponding author.

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
