# Peer review of "Aortic Valve Replacement: Totally Endoscopic versus Mini-Sternotomy"

_jcm, 2023, doi:10.3390/jcm12237300_

Round 1
Reviewer 1 Report
Comments and Suggestions for Authors
I reviewed with interest the manuscript by Yilmaz et al "Aortic valve replacement: totally endoscopic versus mini-sternotomy". In this study, the authors analyzed their center's unique experience in performing minimally invasive aortic valve replacement in two different ways. The article presents a large clinical material (715 patients) almost evenly distributed among the comparison groups. The authors were able to show the advantages and disadvantages of each of the methods used, which may be interesting for clinical practice. However, I have comments and questions that I would like answers to from the authors.
1. An important limitation of the study is that patients were included in different treatment groups at different times. On the one hand, a learning curve could be noted for the totally endoscopic aortic valve replacement technique. On the other hand, performing totally endoscopic aortic valve replacement at a later date could be accompanied by an improvement in the perioperative management of patients in the clinic. Therefore, this important limitation of the study should be noted in the appropriate section of the manuscript.
2. The authors note in section 2.4. Statistical analysis: "Moreover, to correct for baseline differences between the two groups, all possible confounders were determined based on demographic variables that were significantly different between the two groups. These confounders included age, smoking, chronic kidney disease (CKD), AF, arterial hypertension, New York Heart Association (NYHA) score, cardiac family history, preoperative neurological incidents, and the presence of a bicuspid valve." However, in Tables 2-4 I did not see these confounders taken into account when comparing groups. Apparently, it would be more correct to make comparisons in groups after pseudorandomization (propensity score matching).
3. In Figures 2-3 and the authors compare 5-year results in groups. How did they do this when the median follow-up in the TEAVR group was 742.9 days?
4. The writing style of the Discussion section needs correction. You should not start it with references to the literature; it is necessary to cite the main result of the study. Also, you should not re-present your results, but rather compare them with other results.
5. The TEAVR technique is not very widespread, so it is necessary to add links to recent publications in the Discussion (see Ref. 1-4 below)
Refrences^
1. Claessens J, Goris P, Yilmaz A, Van Genechten S, Claes M, Packlé L, Pierson M, Vandenbrande J, Kaya A, Stessel B. Patient-Centred Outcomes after Totally Endoscopic Cardiac Surgery: One-Year Follow-Up. J Clin Med. 2023 Jun 30;12(13):4406. doi: 10.3390/jcm12134406.
2. Zoni D, Cresce GD, Hinna-Danesi T, Benvegnù L, Poddi S, Gallo M, Sella M, Salvador L. Endoscopic aortic valve surgery in isolated and concomitant procedures. Interdiscip Cardiovasc Thorac Surg. 2023 Jun 1;36(6):ivad101. doi: 10.1093/icvts/ivad101.
3. Gu W, Zhou K, Wang Z, Zang X, Guo H, Gao Q, Teng Y, Liu J, He B, Guo H, Huang H. Totally endoscopic aortic valve replacement: Techniques and early results. Front Cardiovasc Med. 2023 Jan 9;9:1106845. doi: 10.3389/fcvm.2022.1106845.
4. Sun J, Yuan Y, Song Y, Hu Y, Bai X, Chen J, Zhong Q. Early results of totally endoscopic robotic aortic valve replacement: analysis of 4 cases. J Cardiothorac Surg. 2022 Jun 13;17(1):155. doi: 10.1186/s13019-022-01899-3.
Comments on the Quality of English Language
No comments
Author Response
Comment 1: We agree with the reviewer and added this to the limitation section of the discussion
Comment 2: We tried propensity score matching but the matching failed. All confounders remained significantly different between the two groups. We tried both nearest neighbour matching and exact matching. Therefore, we were advised by a statistician to correct for the confounders using linear and logistic regression which is explained in the statistical section. We changed the order of the explanation for the correction to make it more clear. In the tables you can see that we added the estimates/odds ratio and confidence intervals of the regressions so confounders are taken into account.
Comment 3: The five year follow-up is done in 72,1% of the Mini-AVR patient but only in 9% of the TEAVR patients. Since there is indeed no 5 year follow up for all patients we added the numbers at risk below the figure to indicate how many patients were analysed at the different timepoints. However, we can remove the 5-year MACCE figure.
Comment 4: We agree with the reviewer that the first sentence where we state the literature may be unnessesary. However, we then explain our primary outcome and compare this to other results. We state our outcome and then what was found in previous studies. Next we elaborate on why we think we found these results like for example why the ICU stay may be longer and the pacemaker rates might be higher in our series. Is your suggestion to leave all the numbers out of the discussion? We added them so it is easier to compare these to results from previous studies.
Comment 5: We agree with the reviewer and added the references except reference 4. This is a study regarding robotic endoscopic AVR which cannot be compared to TEAVR with endoscopic instruments.
Reviewer 2 Report
Comments and Suggestions for Authors
In this article, Drs. Yilmaz and colleagues present a retrospective review of clinical outcomes utilizing two different minimal-access approaches to surgical aortic valve replacement at their institution; partial upper sternotomy ("mini-AVR") and totally endoscopic ("TEAVR"). The authors found several statistically significant differences in outcomes, but no difference in short or longer term mortality.
The article is well written and thoughtfully presented, and their conclusions are sound. Their clinical outcomes are impressive. I do have several comments for the authors:
-The fact that more "sutureless" valves were used in the TEAVR approach (18% vs. 2% in mini-AVR) is of some potential concern. Sutureless surgical aortic valves have not been widely adopted as there is fairly well established evidence that they are associated with greater rates of perivalvular leak (PVL) and higher likelihood of permanent pacemaker (PPM) implantation. It is critical tenet of minimal access surgery that no potential benefits with regards to cosmesis or time to recovery should come at the cost of a compromise in the quality of the operation.
In fact there was a higher need for PPM in the TEAVR group. Can the authors comment on what proportion of PPM were in patients with sutureless valves? In addition, the authors comment in the text about a combined rate of PVL, endocarditis and readmissions during follow-up, but I do not see data (in text or tables) for actual rates of PVL in both groups.
Do the authors feel that the use of many more sutureless valves (with likely inferior outcomes) justifies the potential advantages of TEAVR in terms of less transfusion and shorter length of stay?
-The fact that the TEAVR approach took significantly longer in spite of the use of more sutureless valves (which are faster to implant) suggests that this may be a significantly more challenging technical approach - can the authors comment on this?
-The authors use the term "revision" throughout the manuscript to refer to any return to the operating room. This term is potentially deceptive as it may imply a "revision" was needed of the surgical valve (when in fact most reoperations were not for valve "revision"). It would be more clear, accurate, and conventional to instead use the term "reoperation" in this context.
Comments on the Quality of English Language
Several English language or typographical errors:
-In line 2 under "2.1 Patients," the sentence reads that "The first TEAVR was a fact in October 2017." This should instead read that the first TEAVR "was performed at the study institution in October 2017."
-In the discussion section the authors state that "Revision surgery is always performed endoscopically in TEAVR, which is only done in one of the mini-AVR cases." This sentence is not grammatically correct. I also find it confusing - are the authors implying that for reoperation in one of the mini-AVR cases an endoscopic approach was undertaken? Unless this was a VATS for retained hemothorax, it is not clear to me why one would return a mini-AVR patient to the OR and not utilize the original incision/approach.
-Towards the end of the Discussion section the authors write "Seen our promising results..." This is grammatically incorrect.
Author Response
Comment 1: We agree with the reviewer that sutureless valves are known to have higher PPM and PVL rates. For the PPM this was also the case in our population. In 9 of 12 PPM in the TEAVR group a sutureless valve was implanted. This indicates that that the high PPM rate in the TEAVR group is most likely related to the use of sutureless valve and not the technique itself. We added this in the discusion.
Comment 2: The rate of PVL, endocarditis and cardiac readmissions is mentioned in table 3 in the second part of the table (categorical) on line 6, 7 and 19 . We did write paravalvular reurgitation instead of PVL so we changed that.
Comment 3: The use of sutureless valves is not required to perform TEAVR. We agree that these inferior outcomes should not be accepted to justify the advantages of TEAVR but in our opinion these are linked to the use of sutureless valve and not to the TEAVR technique itself.
Comment 4: We agree with the reviewer that this is technically more challenging and that there will be a learning curve. We elaborated on this in the discussion section but added that it's technically challenging.
Comment 5: We agree with the reviewer and changed revision to reoperation.
Quality of English: We tried to adjust these comments.
Round 2
Reviewer 1 Report
Comments and Suggestions for Authors
The authors answered my questions and made corrections to the text of the manuscript. However, I think it is incorrect to present the MACCE figure for 5 years; the 5-year MACCE figure should be removed.
Comments on the Quality of English Language
No comments
Author Response
We have removed the MACCE figure for 5 years. Should we also remove the 5 year mortality? And the 5 year MACCE data in table 4?
Round 3
Reviewer 1 Report
Comments and Suggestions for Authors
Should we also remove the 5 year mortality? And the 5 year MACCE data in table 4?
- Yes, of course, it is also better to delete this data, since it is not very informative.
Comments on the Quality of English Language
No comments
Author Response
We have removed all the data regarding the 5-year follow-up.